# Functional Connectivity Alterations Based on Hypometabolic Region May Predict Clinical Prognosis of Temporal Lobe Epilepsy: A Simultaneous ^18^F-FDG PET/fMRI Study

**DOI:** 10.3390/biology11081178

**Published:** 2022-08-05

**Authors:** Yi Shan, Hu-Cheng Zhou, Kun Shang, Bi-Xiao Cui, Xiao-Tong Fan, Qi Zhang, Yong-Zhi Shan, Jie-Hui Jiang, Guo-Guang Zhao, Jie Lu

**Affiliations:** 1Department of Radiology and Nuclear Medicine, Xuanwu Hospital, Capital Medical University, Beijing 100053, China; 2Beijing Key Laboratory of Magnetic Resonance Imaging and Brain Informatics, Beijing 100053, China; 3Shanghai Institute for Advanced Communication and Data Science, Shanghai University, Shanghai 200444, China; 4Department of Neurosurgery, Xuanwu Hospital, Capital Medical University, Beijing 100053, China

**Keywords:** temporal lobe epilepsy, simultaneous PET/MR, hypometabolism, functional connectivity, clinical prognosis

## Abstract

**Simple Summary:**

Accurate localization of the epileptogenic zone and understanding the related whole-brain functional alteration patterns are critical for the prediction of postsurgical outcomes in patients with temporal lobe epilepsy (TLE). Previous studies suggested that functional connectivity (FC) alterations based on structurally abnormal lesion on MRI could indicate the clinical prognosis. However, hypometabolic regions localized by ^18^F-FDG PET precedes the appearance of structural abnormalities in lesion localization. By using hybrid ^18^F-FDG PET/fMRI, we aim to obtain spatially and temporally matched structural, functional, and metabolic information of TLE patients, localize their hypometabolic regions, compare the differences of FC alteration patterns based on hypometabolic region and structural lesion, respectively, and further explore their relationships with disease progression and postsurgical outcome. The results showed that hypometabolic region could be used for detecting a more extensive long-range functional alteration pattern, which correlated with disease progression (epilepsy duration) and surgical outcome (seizure-free or not in follow-up). In this way, we conclude that FC alterations based on hypometabolic region constructed by ^18^F-FDG PET/fMRI may provide additional value in the prediction of clinical prognosis in TLE patients.

**Abstract:**

(1) Background: Accurate localization of the epileptogenic zone and understanding the related functional connectivity (FC) alterations are critical for the prediction of clinical prognosis in patients with temporal lobe epilepsy (TLE). We aim to localize the hypometabolic region in TLE patients, compare the differences in FC alterations based on hypometabolic region and structural lesion, respectively, and explore their relationships with clinical prognosis. (2) Methods: Thirty-two TLE patients and 26 controls were recruited. Patients underwent ^18^F-FDG PET/MR scan, surgical treatment, and a 2–3-year follow-up. Visual assessment and voxel-wise analyses were performed to identify hypometabolic regions. ROI-based FC analyses were performed. Relationships between clinical prognosis and FC values were performed by using Pearson correlation analyses and receiver operating characteristic (ROC) analysis. (3) Results: Hypometabolic regions in TLE patients were found in the ipsilateral hippocampus, parahippocampal gyrus, and temporal lobe (*p* < 0.001). Functional alterations based on hypometabolic regions showed a more extensive whole-brain FC reduction. FC values of these regions negatively correlated with epilepsy duration (*p* < 0.05), and the ROC curve of them showed significant accuracy in predicting postsurgical outcome. (4) Conclusions: In TLE patients, FC related with hypometabolic region obtained by PET/fMRI may provide value in the prediction of disease progression and seizure-free outcome.

## 1. Introduction

Early surgical treatment is superior to prolonged medical therapy in patients with temporal lobe epilepsy (TLE), which is the most common type of epilepsy [1]. Accurate localization of the epileptogenic zone (EZ) and understanding the related whole-brain functional alteration patterns are critical for surgical planning and prediction of postsurgical outcomes [2,3,4]. Structural MRI, which is a primary neuroimaging technique in presurgical evaluation, could identify structurally abnormal lesions related to EZ, especially for TLE with hippocampal sclerosis (HS). However, not all structural abnormalities cause seizures. On the other hand, 20–30% of TLE patients have a normal structural appearance on MRI [5,6]. Therefore, additional methods are needed to localize potential EZ and further depict the impaired functional alteration pattern.

^18^F-fluorodeoxyglucose (FDG) PET is directly related to brain metabolism and could thus reveal hypometabolic areas associated with EZ, which has already been used as one of the most reliable methods for precise and noninvasive presurgical lesion localization [7,8]. Previous studies have shown evidence that the dysfunction with hypometabolism precedes the appearance of structural abnormalities in lesion localization [9,10]. For patients with MRI-negative TLE, hypometabolic regions in the temporal lobe on ^18^F-FDG PET showed consistency with their histopathologic findings [11,12]. Moreover, ^18^F-FDG PET has proved to be beneficial for localization in patients with non-concordant electroencephalography (EEG) and neuroimaging findings, as well as those with dual-pathology cases (coexistence of medial temporal and neocortical seizure foci) [5,13]. On the other hand, functional MRI (fMRI) is generally used to characterize whole-brain functional connectivity (FC) alteration patterns related to EZ in patients with epilepsy and could further indicate their clinical prognosis (such as epilepsy duration and surgical outcome) [14,15,16]. However, in most fMRI research, the regional lesion related to EZ was defined by structurally abnormal MRI findings [17,18]. In this way, we hypothesize that FC alteration patterns based on hypometabolic region localized by ^18^F-FDG PET could show potential ability in prognosis of clinical outcome, compared with those based on structural lesions.

To explore this problem, it is essential to simultaneously acquire both brain structural, metabolic, and functional information of patients in the same physiological state. Hybrid ^18^F-FDG PET/MR could provide optimal spatially and temporally matched brain images and thus minimize the physiological variability across subjects [19]. Therefore, by using hybrid ^18^F-FDG PET/fMRI data, we aim to localize the hypometabolic region in patients with TLE, compare the differences in FC alteration patterns based on hypometabolic region on PET and structural lesion on MRI, respectively, and further explore their relationships with disease progression and postsurgical outcome. To achieve this purpose, we recruited specific TLE patients with a homogeneously structural lesion (patients with unilateral HS verified on MRI) and a 2–3-year clinical follow-up after surgical treatment.

## 2. Materials and Methods

### 2.1. Subjects

The study was approved by the Ethics Committee of Xuanwu Hospital, and written informed consent was obtained from all participants before the PET/MR scan. A total of 32 consecutive patients with unilateral TLE-HS who were preoperatively evaluated between April 2017 and May 2018 participated in this study (demographic characteristics of participants see Table 1). In total, 26 age and sex-matched healthy controls were also recruited. The diagnosis of TLE-HS was established with a standardized assessment protocol, including medical history, seizure semiology, neuropsychological examination, EEG surveillance, and MRI. Brain MRI showed unilateral HS (clear morphological atrophy on T1-weighted image and hyperintense signal on T2-fluid attenuated inverted recovery (T2-FLAIR) sequence). Patients with prior neurosurgery were excluded. In total, 24 patients underwent anterior temporal lobectomy (ATL) with pathological results, and the other eight patients underwent radiofrequency thermocoagulation (RFTC). Surgical outcomes were assessed according to the Engel classification [20]. After 2–3 years of follow-up, there were 20 Engel Class IA patients (seizure-free after surgical treatment; ATL treatment: 18; RFTC treatment: 2) and 10 Engel Class non-IA patients (persistent seizures after surgical treatment; ATL treatment: 6; RFTC treatment: 4). Two patients with RFTC were lost to follow-up.

### 2.2. PET/MR Acquisition and Preprocessing

All subjects underwent a hybrid brain PET/MR scan (Signa; GE Healthcare, WI, USA). The scanning protocol was as same as our previous studies [12,21]. Each participant was instructed to fast for at least six hours and received an intravenous injection of 3.7 MBq/kg of ^18^F-FDG. The simultaneous data acquisition was initiated 40 min after the injection. During the scanning, the participants were asked to remain in an awake and relaxed state. The detailed parameters of the PET images were as follows: 8 iterations, 32 subsets, and full width at half-maximum (FWHM) of a Gaussian filter of 3.0 mm. The image matrix was 192 × 192, and the slice thickness was 2.44 mm. Sagittal three-dimensional brain volume T1-weighted images were acquired with a repetition time/echo time of 8.5/3.2, flip angle of 15°, and voxel size of 1.0 × 1.0 × 1.0 mm^3^. Functional MRI data were collected using a single-shot echo-planar imaging sequence with parameters as follows: repetition time/echo time, 2000/30; flip angle, 90°; gap, 0.8 mm; voxel size, 3.0 × 3.0 × 3.0 mm^3^; slice number, 33; and volume number, 300.

PET data of all subjects were preprocessed using SPM8 (Wellcome Department of Clinical Neurology, London, UK). First, the PET scan for each subject was registered with a corresponding T1-weighted MRI scan. Second, MRI images were segmented into gray matter (GM), white matter, and cerebrospinal fluid tissue probability maps using the unified segmentation method. Then, the GM map was registered to the Montreal Neurological Institute (MNI) stereotaxic template by using the forward parameters estimated during the unified segmentation. The registered PET image was also normalized to the MNI template using the same transformation parameters. Finally, the normalized GM and PET images were smoothed equivalent to a convolution with an isotropic Gaussian kernel of 6 mm to increase signal-to-noise ratios.

fMRI images were preprocessed using the Data Processing Assistant for Brain Imaging (DPABI_V4.1; http://www.rfmri.org/dpabi (accessed on 25 July 2019)). The first 10 volumes were removed for image stabilization and the participant’s adaptation to the scanning. First, the remaining volumes were corrected for time differences between slices and then realigned to the first volume for motion correction. Next, the structural image was coregistered to the average functional image. The linear trend of time courses of functional images was removed, and nuisance signals (including Friston 24-head motion parameters, white matter, and cerebrospinal fluid) were extracted and regressed out from the data. Then, the transformed structural images were segmented into GM, white matter, and cerebrospinal fluid. The functional volumes were spatially normalized to MNI standard space with the normalization parameters from unified segmentation. The resulting images further underwent spatial smoothing with a Gaussian kernel of 6 mm FWHM. Finally, bandpass filtering (0.01–0.1 Hz) was performed on the time courses for FC analysis. Subjects were excluded if the maximal displacement was >3 mm or the maximal rotation was >3° in any direction.

### 2.3. Obtaining Hypometabolic Regions on ^18^F-FDG PET

We used a two-step approach to localize the FDG-hypometabolism as the region of interest (ROI). First, the PET images of all patients were visually assessed in a blind mode by two radiologists individually without any knowledge of clinical or pathological results. They were asked to identify the brain regions with hypometabolism. Any disagreement between the two observers was resolved by consulting a third radiologist to reach a final consensus. Second, these PET images were scaled to the global mean activity in a whole-brain mask to report relative regional glucose metabolism (also called standardized uptake value ratio maps, SUVr maps). In this step, we performed a voxel-wise two-sample *t*-test to compare the group differences in SUVr maps between patients and the controls. The probabilistic GM maps together with age and sex were included as group scalar covariates. Statistical t-maps were further corrected for false-positive clusters, applying the false discovery rate (FDR) correction (corrected *p* < 0.001; cluster size ≥50 voxels). The results of the two steps were compared. Because both results were agreed, we used voxel-wise-based results as ROI for further analysis.

The gray matter volume (GMV) of these hypometabolic brain areas was also obtained by calculating the average value of the GM maps in the masks of corresponding hypometabolic brain areas from the automated anatomical labeling atlas. The GM maps and GMV values between patients and controls were compared, and the correlations between the GMV and SUVr in patients were obtained. A whole brain voxel-wise two-sample *t*-test was also conducted to compare the group differences in GM maps between patients and the controls.

### 2.4. Construction of ROI-Based FC Alteration Patterns

Whole-brain FC alteration patterns based on hypometabolic region (ROI obtained from the above step) and structural lesion (defined as the affected hippocampus) as two ROIs were analyzed, respectively. Individual FC maps of each ROI were generated by calculating Pearson’s correlation coefficients between the mean time course of ROI and the time course of each voxel in the whole-brain. Subject-level correlation maps were converted to z-value maps with Fisher’s transformation to improve the normality. Then, we performed a voxel-wise two-sample *t*-test to compare the group differences between FC maps in patients and controls. The probabilistic GM maps, age, and sex were included as group scalar covariates. We considered the results to be significant at voxel *p* < 0.01 and cluster *p* < 0.05, with Gaussian random field (GRF) corrections for multiple comparisons. The FC results based on the two ROIs were compared, and the brain areas only showing significantly changed connections with hypometabolic region were identified. The FC values between the ROI and brain regions with significant changes were extracted.

### 2.5. Statistical Analysis

Demographic and clinical characteristics were compared by analysis of variance, Chi-square tests, and two-sample *t*-tests as appropriate. The relationships between SUVr and GMV, as well as the relationship between FC values and clinical parameters, were assessed by using Pearson’s correlation analyses. Receiver operating characteristic (ROC) curve analysis was used to assess the classification ability of two ROIs in distinguishing patients from controls, respectively. ROC curve analysis based on FC values between the two ROIs and significantly changed brain regions were used to predict surgical prognosis. The highest area under the curve (AUC), sensitivity, and specificity were obtained. The level of significance was set at *p* < 0.05. Statistical analyses were performed using SPSS (v.22, IBM, Armonk, NY, USA) and R statistical software (2014, R Core Team, Vienna, Austria).

## 3. Results

### 3.1. Demographic and Clinical Characteristics of Subjects

Table 1 shows the demographic information of all subjects, including 14 patients with left TLE (6 female; ages: 25 ± 15.3 years), 18 patients with right TLE (9 female; ages: 28.1 ± 6.4 years), and 26 controls (14 female; ages: 31.7 ± 6.8 years). A significant difference was found in age between patients with left TLE and controls (*p* < 0.05). No significant difference was found in age between patients with right TLE and controls (*p* > 0.05). No significant difference in sex, age of onset, epilepsy duration, seizure frequency, type of surgery, or Engel classification was found between patients with left and right TLE (*p* > 0.05).

### 3.2. Hypometabolic Region Localized on ^18^F-FDG PET

In the visual assessment, hypometabolic regions located at the ipsilateral temporal lobe and hippocampus were observed on PET images in all patients. After ATL, the pathological results of the affected hippocampus showed HS type I, and the ipsilateral temporal lobes showed focal cortical dysplasia (FCD) or gliosis. These findings were consistent with a preoperative visual assessment of the FDG-hypometabolism (see Figure 1 for a typical case).

The voxel-wise differences between patients and controls in FDG-PET images are shown in Figure 2A and Table 2. Compared with the controls, patients with left TLE showed significantly decreased brain metabolism in the left temporal lobe, involving the hippocampus, parahippocampal gyrus, superior temporal gyrus, middle temporal gyrus, and inferior temporal gyrus (FDR corrected, *p* < 0.001). Similar results were found in patients with right TLE: hypometabolic regions were localized in the right hippocampus, parahippocampal gyrus, superior temporal gyrus, middle temporal gyrus, and inferior temporal gyrus (FDR corrected, *p* < 0.001). Compared to visual assessment, consistent focal hypometabolic brain areas were found in the voxel-wise analysis. The cluster showing a significant SUVr decrease in the left TLE group was larger than that in the right TLE group.

Then, we compared the GMV of the above hypometabolic areas identified by voxel-wise analysis to determine whether they had structural atrophy. Significantly decreased GMV in the ipsilateral hippocampus was observed in both left and right TLE (left: *p* < 0.001; right: *p* = 0.035; Figure 2B), which showed consistency with the results from voxel-wise analysis (Appendix A). However, there were no significant differences in the parahippocampal gyrus, superior temporal gyrus, middle temporal gyrus, or inferior temporal gyrus between patients and controls. In addition, there were no relationships between the SUVr and GMV value in these five hypometabolic regions (*p* > 0.05, Appendix A).

### 3.3. Comparisons of FC Alteration Patterns Based on Hypometabolic and Structural Lesion

Comparisons of whole-brain FC alterations between TLE patients and controls are shown in Figure 3A and summarized in Table 3. In general, functional alteration patterns based on hypometabolic region showed a more extensive long-range FC reduction compared with those based on structural lesion. In patients with left TLE, brain areas showing reduced FC with the left hippocampus (structural lesion) were located in the posterior cingulate gyrus and precuneus (posterior default mode network (DMN)), right middle frontal gyrus, and temporal lobe (GRF corrected, *p* < 0.01). In patients with right TLE, brain areas showing reduced FC with the right hippocampus (structural lesion) were located in the posterior DMN, bilateral middle frontal gyri, bilateral temporal gyri, and cerebellum (GRF corrected, *p* < 0.01). Concerning the FC alterations with hypometabolic region, patients with left TLE showed diminished FC in brain areas located in the posterior DMN, bilateral frontal lobes, medial frontal areas, sensorimotor cortex, bilateral temporal lobes, left occipital lobe, and cerebellum (GRF corrected, *p* < 0.01). In patients with right TLE, brain areas showing reduced FC with right hypometabolic region were located in the posterior DMN, bilateral frontal lobes, medial frontal areas, bilateral temporal lobes, and cerebellum (GRF corrected, *p* < 0.01). Regardless of the lesion side, brain areas only showing significantly decreased FC with hypometabolic lesion were located in bilateral medial frontal gyri and middle temporal gyri, left superior frontal gyrus, and right superior temporal gyrus. The ROC curve of FC values based on hypometabolic ROI was slightly superior to that based on structural ROI in classifying TLE patients from controls (AUC: 0.89 vs. 0.86, Figure 3B).

### 3.4. Correlations between FC Alterations Based on Hypometabolic Region and Clinical Prognosis

Brain areas showing significantly decreased FC with the two ROI were used to evaluate the correlation between functional alteration patterns and clinical prognosis (epilepsy duration and surgical outcome of the patients). For hypometabolic ROI, when considering them as a whole region, no significant relationship was found between the averaged FC value and the time of epilepsy duration (*p* > 0.05). However, when considering the brain areas separately, the values of FC between bilateral medial frontal gyri and hypometabolic ROI were negatively correlated with the time of epilepsy duration in TLE patients (left: r = −0.41, *p* = 0.02; right: r = −0.38, *p* = 0.03; uncorrected; Figure 4A,B), while those of other brain areas did not show significant correlation (*p* > 0.05, Figure 4C–F). For structural ROI, no significant relationship was found between the FC values and the time of epilepsy duration (Appendix A). ROC curves were constructed to evaluate the classification ability of decreased FC values related to the two ROIs in the prediction of surgical outcome (Engel Class IA or non-IA). When considering all brain areas showing decreased FC with hypometabolic ROI as a whole region, the averaged FC value showed significant classification ability with an AUC of 0.630, a sensitivity of 50%, and a specificity of 83.3% (Figure 5). However, better performance was found when only considering the above six brain regions showing decreased FC with hypometabolic ROI but normal FC with structural ROI, with an AUC of 0.833, a sensitivity of 66.7%, and a specificity of 100%. When considering these brain areas separately, valuable ROC curves were found in the FC values between hypometabolic ROI and left middle temporal gyrus, right superior temporal gyrus, left superior frontal gyrus, with an AUC of 0.731, 0.694, 0.639, respectively. The AUC of FC values based on structural ROI was 0.530.

## 4. Discussion

In this study, we localized the hypometabolic region in patients with TLE, compared the differences of FC alteration patterns based on hypometabolic region and structural lesion, respectively, and further explored their relationships with disease progression and postsurgical outcome. We found that visual assessment combined with voxel-wise analysis could identify the hypometabolic region of TLE patients, which was consistent with their pathological results. Hypometabolic region could be used to detect more extensive FC alteration patterns correlated with disease progression and surgical outcome, which may reflect a valuable epileptogenic network in TLE patients. To the best of our knowledge, our study with hybrid ^18^F-FDG PET/fMRI may be the first to report the potential value of hypometabolic region in depicting impaired brain function and their relationship with clinical prognosis, taking advantage of the simultaneously acquired structural, metabolic and functional information in the same pathophysiological states of patients.

For the localization of the hypometabolic region, visual assessment has been widely used in clinical practice, and the accuracy could reach up to 85–90% in patients with TLE [5]. It has been reported that ^18^F-FDG PET based on visual assessment was more sensitive than structural MRI, especially in patients with FCD [22,23]. The visual assessment results of our study were mainly located at the ipsilateral hippocampus and temporal lobe with hypometabolism, which showed concordance with histopathological results (hippocampus with HS type I and temporal lobe with FCD). However, visual assessment is subjective and could not be used for further quantitative analysis between groups. Hence, we used voxel-wise analyses to further verify the localization of the hypometabolic region, and the results were consistent with those in the visual assessment, which were also consistent with those of previous studies [24,25,26,27]. For example, Chassoux et al. reported that focal anteromesial temporal hypometabolism was associated with Engel Class IA outcomes [27]. In our patients, we also found focal hypometabolism in the ipsilateral temporal lobe and hippocampus, and 75% of them showed Engel Class IA after ATL. On the other hand, although the hypometabolic areas were more restricted to the ipsilateral temporal lobe, patients with left TLE in our study showed more extensive metabolic changes than those with right TLE. This finding supports previous evidence reporting that the epileptogenic network could be asymmetrical and depend on the lateralization of TLE patients [28,29,30].

We further evaluated the extent of structural atrophy in hypometabolic areas and found significant GMV reduction only in the affected hippocampus, which was consistent with previous studies [31,32]. Moreover, no relationship was found between decreased FDG uptake and GM loss. These findings can be supported by previous viewpoints that the degree of hippocampal atrophy or associated temporal atrophy did not appear to be determinant for metabolic changes, and, thus, the localization of hypometabolic changes may precede identifying structural atrophy in TLE patients [10,25].

For depicting functional alteration patterns of TLE patients, previous studies usually selected the sclerotic hippocampus as ROI [15,17,18]. In our study, we also evaluated the FC alteration pattern based on hippocampus as structural ROI, and the results were, in part, consistent with previous studies, showing reduced long-range FC from the affected hippocampus to specific brain areas such as the posterior DMN [15,18]. However, we compared the FC alteration pattern based on hypometabolic ROI and structural ROI, respectively and found that the former one revealed a more extensive long-range FC reduction in the whole-brain. The ROC curve of reduced FC based on hypometabolic ROI was also slightly superior based on structural lesion in classifying TLE patients from controls. Notably, brain areas only showing reduced FC with hypometabolic ROI (e.g., medial frontal gyrus) were found to be negatively related to disease progression. It has been reported that decreased FC in the frontal cortex is related to the various conditions of neurocognitive impairments [33,34]. In this way, our findings indicate that longer epilepsy duration could contribute to possible psychiatric and cognitive symptoms.

Here, it is worth mentioning that using hypometabolic ROI determined by voxel-wise differences between groups in FC and subsequent analysis will ignore the individual variability across patients. However, defining a constant ROI for all subjects in the ROI-based FC analysis is also important, because setting different ROI for each subject could make a big difference to the results of FC calculation. In our study, we chose to use a constant hypometabolic ROI for all subjects, but, meanwhile, we made efforts to improve the homogeneity by selecting restricted patients. For all of our 32 patients, in the visual assessment, although the hypometabolic lesions of every patient were not exactly the same, they presented a rather similar appearance of hypometabolism in the ipsilateral temporal lobe. In the voxel-wise analysis, the hypometabolic regions in TLE patients were restricted to the ipsilateral temporal lobe, without any extratemporal, contralateral or hypermetabolic lesions. These findings indicated that the enrolled cases in our study had high homogeneity in terms of FDG-hypometabolism. Therefore, we consider that the influence caused by the individual variability of hypometabolic regions across patients has been substantially controlled.

Surgical treatment is superior to prolonged medical therapy in patients with TLE; however, a proportion of them continue having seizures until 2 years after surgery [35]. Several preoperative neuroimaging methods have been used to predict the postsurgical outcomes, focusing on the construction of accurate functional, structural and metabolic alteration patterns [27,36]. In our study, FC alterations related to hypometabolic region showed significant ability in classifying surgical outcome (Engel Class IA or non-IA) of TLE patients, and the accuracy of prediction was even higher in brain regions only showing decreased FC with hypometabolic region but normal FC with structural lesion. These results verified our hypothesis that when combining ^18^F-FDG PET and fMRI data, FC alteration based on hypometabolic region could provide additional value in the prediction of surgical prognosis in TLE patients.

There were several limitations in this study. First, the difference in sample size and outcome between the two different surgical treatments (ATL and RFTC) may lead to a mixed effect, therefore reducing the support for the conclusion that FC to hypometabolic regions is associated with outcome. A larger sample size with unified surgical treatment is needed in future studies. Additionally, because patients with right and left TLE represent two different entities, the two subgroups should be considered separately but the number of each group was relatively small. The age of patients with left TLE and controls were slightly mismatched. Although we included age as a covariate in the statistical analysis to minimize the possible influence, future studies with larger cohorts are needed to further confirm our findings. Second, long-term epileptic activities can contribute to cognitive symptoms, but we lacked additional neuropsychological measures in our clinical follow-up. Verification on the relationship between FC alterations and cognitive changes could be applied to further confirm our findings.

## 5. Conclusions

In this study, we focused on FC alterations based on hypometabolic region obtained by ^18^F-FDG PET/fMRI in patients with TLE-HS. Compared with structural lesion, hypometabolic region could be used for detecting a more extensive long-range functional alteration pattern, which showed an association with disease progression and seizure-free outcome. Our findings may provide additional value in the prediction of clinical outcomes in TLE patients.

## Figures and Tables

**Figure 1 biology-11-01178-f001:**
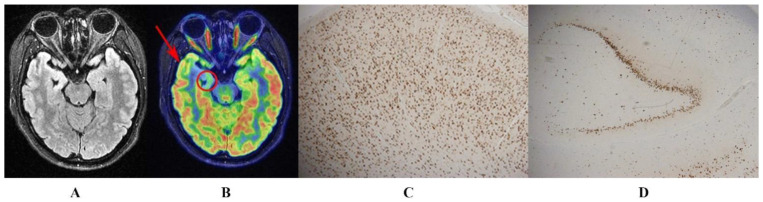
A 21-year-old woman with a history of seizures, onset at age 15. T2-FLAIR showed right hippocampal atrophy with a hyperintense signal (**A**). Hybrid ^18^F-FDG PET/MR image indicated a well-defined area of focal hypometabolism in the right temporal lobe (arrow) and hippocampus (circle) (**B**). After a right anterior temporal lobectomy, the histopathological images with neuronal nuclear antigen immunostaining showed FCD type Ib in the right temporal lobe (**C**), and HS type I in the ipsilateral hippocampus (**D**). After a postoperative 2-year follow-up, the patient was classified as Engel Class IA.

**Figure 2 biology-11-01178-f002:**
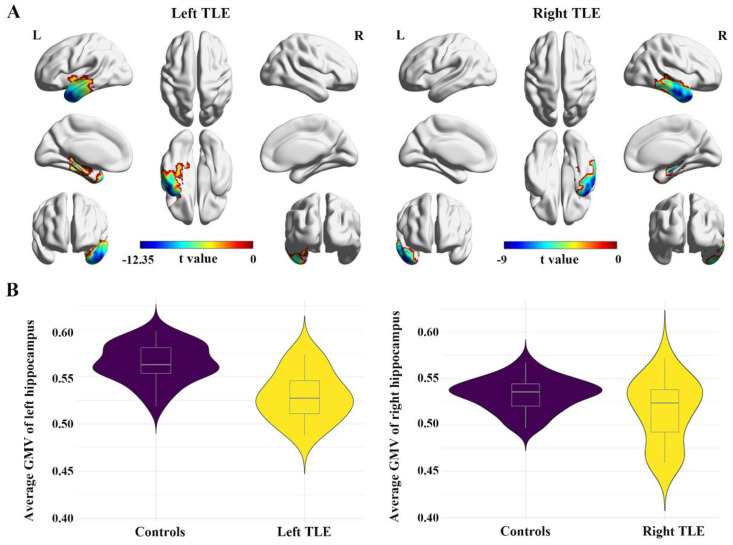
Localization and the GMV of hypometabolic brain regions in TLE patients compared with controls. Significantly decreased brain metabolism was found in brain areas of the ipsilateral hippocampus, parahippocampal gyrus, and superior, middle, and inferior temporal gyri using the voxel-wise comparison between groups ((**A**), FDR corrected, *p* < 0.001). Significantly decreased GMV was found only in the ipsilateral hippocampus ((**B**), *p* < 0.05).

**Figure 3 biology-11-01178-f003:**
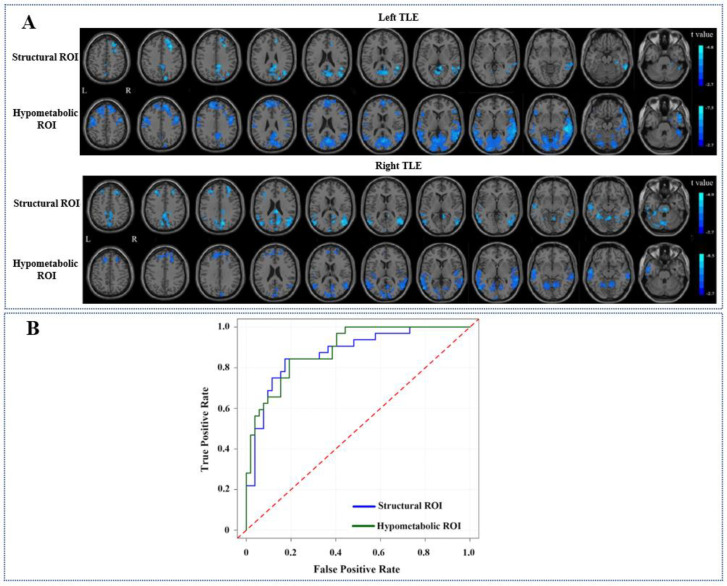
Whole-brain ROI-based FC alterations between TLE patients and controls. Brain areas with cold color indicate decreased FC with structural or hypometabolic ROI, respectively ((**A**), GRF corrected, *p* < 0.01). ROC curves were constructed to assess the ability of FC values related to hypometabolic and structural ROI, respectively, in classifying TLE patients from controls (**B**).

**Figure 4 biology-11-01178-f004:**
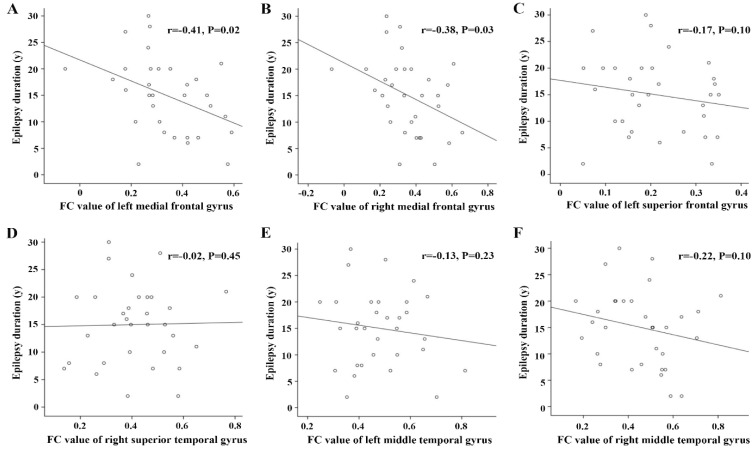
Correlations between epilepsy duration and FC values of brain areas showing decreased connection with hypometabolic ROI. FC values of left (**A**) and right (**B**) medial frontal gyri negatively correlated with the year of epilepsy duration in patients with TLE. There were no relationships between the FC values of other brain regions and epilepsy duration (**C**–**F**).

**Figure 5 biology-11-01178-f005:**
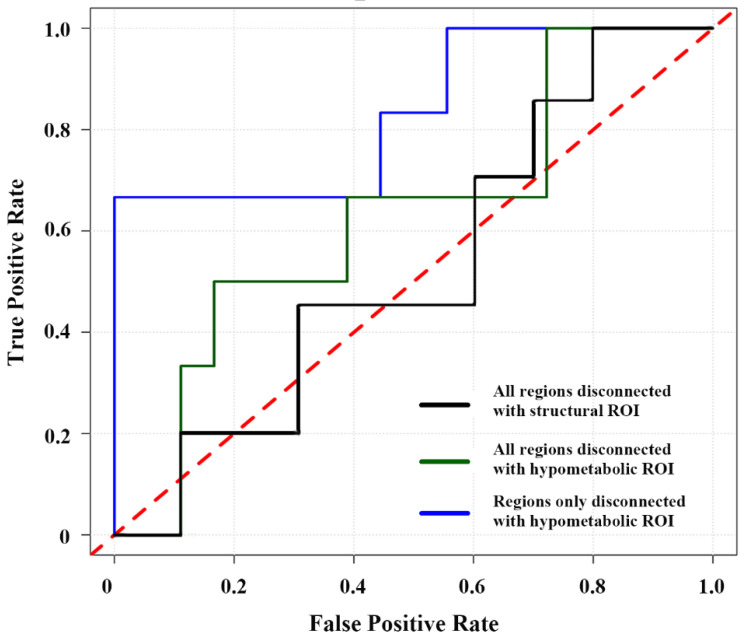
ROC curve evaluating the ability of FC values of brain areas reflecting decreased connection with hypometabolic or structural ROI in the prediction of surgical outcome.

**Table 1 biology-11-01178-t001:** Demographic and clinical characteristics of participants.

	Left TLE	Right TLE	Controls
Number	14	18	26
Gender (male: female)	8:6	9:9	12:14
Age (y, mean ± SD)	25.1 ± 5.3 ^1^	28.1 ± 6.4	31.7 ± 6.8
Age of onset (y, mean ± SD)	11.6 ± 7.0	14.4 ± 9.7	——
Epilepsy duration (y, mean ± SD)	15.5 ± 7.8	14.6 ± 7.0	——
Seizures per month (mean ± SD)	4.6 ± 7.7	5.8 ± 6.7	——
Type of surgery (ATL/RFTC)	10/4	14/4	——
Engel class (IA/non-IA)			
ATL	18/6		——
RFTC	2/4		——

^1^ Compared with controls, *p* < 0.05. ATL, anterior temporal lobectomy; RFTC, radiofrequency thermocoagulation.

**Table 2 biology-11-01178-t002:** Statistical maps and spatial coordinates (MNI space) of brain areas with metabolic alterations in patients compared with controls.

Patients	Metabolic Changes	Brain Areas	Cluster Extend	Peak Voxel			
				t	x	y	z
Left TLE	Hypometabolism	L STG L MTG L ITG L PHG L HG	1629	−12.35	−48	0	−33
Right TLE	Hypometabolism	R STG R MTG R ITG R PHG R HG	1002	−9.00	30	−15	−21

TLE: temporal lobe epilepsy, L: left, R: right, STG: superior temporal gyrus, MTG: middle temporal gyrus, ITG: inferior temporal gyrus, PHG: parahippocampal gyrus, HG: hippocampal gyrus.

**Table 3 biology-11-01178-t003:** Statistical maps and spatial coordinates (MNI space) of brain areas with functional alterations based on different ROIs in patients compared with controls.

TLE	ROI	FC Values	Brain Areas	Cluster Extend	Peak Voxel			
					t	x	y	z
Left TLE	L-hippocampus	Decreased	PCG/PreCu	808	−4.53	6	−48	9
		Decreased	R MEG/R ITG	603	−4.73	48	−48	15
		Decreased	R SFG/R MiFG	491	−4.83	30	24	42
	Hypometabolic regions	Decreased	B STG/B MTG/B ITG/ B MOG/B IOG/PCG/ PreCu/Cereb	7792	−7.53	54	−36	−6
		Decreased	B SFG/B MiFG/B MeFG/ B OFG/B PoCG/B PrCG	1752	−5.03	−18	54	30
Right TLE	R-hippocampus	Decreased	PCG/PreCu	2111	−4.89	6	−30	24
		Decreased	R MTG	661	−4.45	45	−75	21
		Decreased	L MTG/L ITG	271	−3.89	−45	−21	−33
		Decreased	R SFG/R MiFG	264	−3.68	27	51	33
		Decreased	L SFG/L MiFG	200	−4.10	−30	36	39
	Hypometabolic regions	Decreased	L STG/L MTG/L ITG	1246	−8.54	−54	−12	−21
		Decreased	PCG/PreCu	956	−4.8	6	−57	−9
		Decreased	R STG/R MTG	801	−6.02	66	−39	3
		Decreased	B SFG/B MeFG/B MiFG	563	−4.58	−9	45	30

TLE: temporal lobe epilepsy, L: left, R: right, B: bilateral, Cereb: cerebellum, HG: hippocampal gyrus, ITG: inferior temporal gyrus, IOG: inferior occipital gyrus, MeFG: medial frontal gyrus, MiFG: middle frontal gyrus, MTG: middle temporal gyrus, MOG: middle occipital gyrus, OFG: orbital frontal gyrus, PCG: posterior cingulate gyrus, PHG: parahippocampal gyrus, PoCG: postcentral gyrus, PrCG: precentral gyrus, PreCu: precuneus, SFG: superior frontal gyrus, STG: superior temporal gyrus.

## Data Availability

The datasets used or analyzed during the current study are available from the corresponding author on reasonable request.

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
