# Peer review of "Functional Connectivity Alterations Based on Hypometabolic Region May Predict Clinical Prognosis of Temporal Lobe Epilepsy: A Simultaneous 18F-FDG PET/fMRI Study"

_biology, 2022, doi:10.3390/biology11081178_

Round 1

Reviewer 1 Report

I read this manuscript entitled "Functional Connectivity Alterations Based on Hypometabolic Epileptogenic Zone May Predict Clinical Prognosis of Temporal Lobe Epilepsy: A Simultaneous 18F-FDG PET/fMRI Study" with a great enthusiasm.

In my opinion, a major revision is necessary if the authors will follow my advise.

"Epileptogenic zone" is a conceptual definition, embedding all of data deriving from routine EEG, sleep deprivation EEG, 24-hour scalp  VEEG-monitoring including interictal as well as ictal activity, hetero- and autoanamnesis, ictal and interictal SPECT, invasive EEG recording, cranial MRI, MEG, FDG-PET/CT and if available, FDG-PET/MRI. Data derived from FDG-PET/CT or FDG-PET/MR can be only a part - which is of course a significant part - of the epileptogenic zone. Thus, such expressions like "hypometabolic epileptogenic zone" or "structural epileptogenic zone" are misleading and as itself cannot be helpful for the epileptologist, since if we would interpret "hypometabolic epileptogenic zone" seriously, we should indicate a more extended resective line as in fact we do.

I suggest to introduce an expression: "epileptogenic lesion" for defining e.g. hippocampal sclerosis in the cranial MRI in the lesional cases.

In the case of FDG-hypometabolism, the situation is more complicated, because it can be for example a bilateral temporal hypometabolism, while we operate only one hemisphere, because epileptogenic zone is only in one of the hemispheres, not exluding the opportunity of bilateral FDG-hypometabolism.

Thus I recommend to solve this basic problem and then please reword the complete manuscript, including also the title, avoiding these expressions as "hypometabolic EZ" or "structural EZ".

After the resubmission, I will take care at the minor issues.

Reviewer 2 Report

Reading the manuscript written by Shan et al., was really interesting. The article provided insights functional connectivity alterations based on hypometabolic epileptogenic zone that may predict clinical prognosis of temporal lobe epilepsy in a simultaneous 18F-FDG PET/fMRI study. The results showed that hypometabolic EZ could be used for detecting a more extensive long-range functional alteration pattern, which correlated with disease progression (epilepsy duration) and surgical outcome (seizure-free or not in follow-up). Moreover, they conclude that FC alterations based on hypometabolic EZ constructed by 18F-FDG PET/fMRI may provide additional value in the prediction of clinical prognosis in TLE patients.

Overall, this paper is written in professional English with sufficient introduction, detailed methods and solid data. The article is easy to read, well designated and presented, and can be of interest to reader and researchers.

Author Response

Dear reviewer,

Thank you for your kind comments on our manuscript. We really appreciate your recognition and encouragement.

Best Regards.

Sincerely,

Jie Lu, MD, PhD

Department of Radiology and Nuclear Medicine, Xuanwu Hospital, Capital Medical

University, Beijing 100053, China

Tel: +86-010-83198379

Fax: +86-010-83198379

E-mail: imaginglu@hotmail.com

Reviewer 3 Report

The objective of this work was to test the hypothesis that functional alteration patterns based on the hypometabolic pattern epileptogenic zone using 18FDGPET could show potential ability in prognosis of clinical outcome compared with the epileptogenic zone based on structural abnormality. To explore this hypothesis a cohort of patients with unilateral hippocampal sclerosis on MRI were examined. Two different epileptogenic zones (EZ) were identified – the hypometabolic one which was a set of regions found by group difference in PET signal between patients and controls, and the structural one which was assumed as only the hippocampus. Then the functional connectivity to each of these EZs was examined and compared to duration of disease and outcome. While the question is potentially interesting, the logic of the methods is somewhat confusing.

Hypometabolic differences (hypometabolic EZ) were determined by voxel-wise differences between the patient and controls as a group. This is assumed to be constant across all patients, but this assumption may not be true given the variability in outcome across the patients. Even a different set of patients might have slightly different results from the group analysis, especially due to the relatively small number in each right and left group. There was also a visual assessment. Did every patient show the same regions as hypometabolic? The conclusion that can be drawn from this analysis is that connectivity to this larger set of areas has different characteristics than connectivity to the hippocampus alone. But it is not clear that this set of regions is the hypometabolic EZ in all the individual patients. For example, is the EZ is the same even in the subjects with unfavorable outcome and the connectivity is different? Or is the EZ itself different? Or both?

A voxel-wise analysis should also be performed for the gray matter volume to determine differences between patients and controls, but it not clear if this was performed.

To compare outcome classification between hypometabolism and structure, the ROC for the connectivity to the structural ROI would need to be computed as well. Was this performed? Even if the regions are a subset of the regions of decreased connectivity to the hypometabolic EZ, their connectivity values to the hippocampus alone would presumably be different than their connectivity to all the hypometabolic EZ. Similar to analysis in Figure 3B.

For the correlation between FC connectivity to hypometabolism and duration, there are no relationships that meet significance with correction for multiple comparisons.

Was correlation with duration performed using the regions connectivity to the structural EZ?

There is an important limitation with the two different types of surgery that seem to have very different ratio of patients with seizure free outcome.

Figure 3A – the legend is confusing. I do not understand what is shown on this figure. Do regions of blue and yellow overlap?

Table 1 - There are only 30 patients with ATL or RFTC outcomes in the text (section 2.1) and in Table 1, but there are 32 patients. Please define abbreviations ATL and RFTC in table legend.

Does FDG have an effect on the BOLD signal?

Round 2

Reviewer 3 Report

I appreciate that the authors addressed some of my concerns. However, I remain concerned with the confound of the effect of type of surgery on outcome. This association is almost significant with p=0.053 using the Pearson Chi-Square Test. In addition, by just assuming all ATL are Engel 1A and all RFTC are greater than Engel 1A you get an accuracy of 73% which is greater than using all the hypometabolic regions. Even if this were added to the limitations, this significantly reduces support for the main conclusion of the paper that FC to (some) hypometabolic regions is associated with outcome. This can only be mitigated by more patients or using surgeries where this strong bias in outcome does not exist.

This confound, coupled with the lack of corrected statistical findings in Figure 4, leaves few clearly supported findings.

Some other clarification also remain incomplete:

1.       Why are the two patients without available outcomes included in this work if the point is to relate the FC to the outcome?

2.       Why not plot the information from Figure S2 on Figure 5 for clarity and ease of comparison?

3.       The authors provide a lot of information in their response as to why they believe that the PET hypometabolism is consistent across patients. This should be explicitly stated in the discussion/limitations section of the paper.

Round 3

Reviewer 3 Report

All of my minor concerns have been addressed and the study is described clearly. My primary concerns remain the same. The findings of the PET regions being associated with disease progression and outcome are not sufficiently supported.

Author Response

Dear reviewer,

On behalf of my co-authors, we thank you again for your valuable comments which helped us a lot to improve our study and future research.

Sincerely,

Jie Lu, MD, PhD